# Somatic Mobilization: High Somatic Insertion Rate of *mariner* Transposable Element in *Drosophila simulans*

**DOI:** 10.3390/insects13050454

**Published:** 2022-05-12

**Authors:** Mariana Cancian, Tiago Minuzzi Freire da Fontoura Gomes, Elgion Lucio Silva Loreto

**Affiliations:** 1PPG Genetic and Molecular Biology, Federal University of Rio Grande do Sul (UFRGS), Porto Alegre 90035-003, Brazil; marianacancian1997@gmail.com (M.C.); tiago.minuzzi87@gmail.com (T.M.F.d.F.G.); 2Biochemical and Molecular Biology Department, Universidade Federal de Santa Maria (UFSM), Av. Roraima 1000, Santa Maria 97105-900, Brazil

**Keywords:** *mariner*, *Drosophila*, insertion, somatic transposition

## Abstract

**Simple Summary:**

Transposable elements are capable of promoting somatic mobilization, and this may potentially have biological consequences. However, the rate of somatic mobilization is poorly known. The *mariner* element causes somatic mobilization in *Drosophila simulans*. We used an assay based on sequencing of the flanking *mariner* region and developed a pipeline to identify new *mariner* insertions in *D. simulans*. We found that flies carrying two *mariner* copies (one autonomous and one non-autonomous) had approximately 300 new somatic insertions during fly development (eggs to adult). We show that the rate of somatic insertion of the *mariner* element in *D. simulans* is high.

**Abstract:**

Although transposable elements (TEs) are usually silent in somatic tissues, they are sometimes mobilized in the soma and can potentially have biological consequences. The *mariner* element is one of the TEs involved in somatic mobilization (SM) in *Drosophila* and has a high rate of somatic excision. It is also known that temperature is an important factor in the increase of the *mariner* element SM in the fly. However, it is important to emphasize that excision is only one step of TE transposition, and the final step in this process is insertion. In the present study, we used an assay based on sequencing of the *mariner* flanking region and developed a pipeline to identify novel *mariner* insertions in *Drosophila simulans* at 20 and 28 °C. We found that flies carrying two *mariner* copies (one autonomous and one non-autonomous) had an average of 236.4 (±99.3) to 279 (±107.7) new somatic insertions at 20 °C and an average of 172.7 (±95.3) to 252.6 (±67.3) at 28 °C. In addition, we detected fragments containing *mariner* and others without *mariner* in the same regions with low-coverage long-read sequencing, indicating the process of excision and insertion. In conclusion, a low number of autonomous copies of the *mariner* transposon can promote a high rate of new somatic insertions during the developmental stages of *Drosophila*. Additionally, the developed method seems to be sensitive and adequate for the verification and estimation of somatic insertion.

## 1. Introduction

Transposable elements (TEs) are DNA sequences that can change their position in the genome [1]. The mobilome, the fraction of the genome composed of TEs, varies across species, ranging from as little as 5.6% in *Anopheles darlingi* or 20% in *Drosophila*, to 48% in humans and about 80% in maize [2,3,4,5,6,7]. The distribution of autonomous, non-autonomous, and degenerate copies of TEs in the mobilome of different species is also variable. Autonomous TEs are those capable of producing the enzymes required for transposition, and non-autonomous TEs utilize enzymes produced by autonomous TEs, whereas degenerate copies can no longer transpose [1,8].

Transposition events can have biological consequences. When they occur in germ cells, they can generate genetic variability that can drive evolution. When they occur in somatic tissues, they can sometimes generate beneficial genetic variability, but they can also have adverse effects on organisms [9]. To minimize the deleterious effects, various control mechanisms have evolved, such as DNA methylation, chromatin packing changes [10,11,12], and various mechanisms involving RNA interference. The short-interfering RNAs (siRNA), which are mainly involved in silencing TEs in somatic cells, and PIWI-interacting RNAs (piRNA) are also involved in silencing TEs in germ cells [11,13,14].

Until recently, interest in somatic transposition (ST) was low because it was considered a rare phenomenon due to the existence of control mechanisms [15,16]. However, recent studies have suggested that ST is a common event in many organisms [17,18] and has the potential to be involved in some human diseases. ST may be involved in age-related diseases such as cancer [19,20,21]. Nevertheless, there are several unanswered questions about ST. For example: How frequently does ST occur? What factors enable TEs to escape silencing mechanisms? What are the biological consequences of ST?

*Drosophila* is one of the leading models for studying TEs, as it has a well-annotated and characterized mobilome and has been used as a model organism for over a century [22]. Within *Drosophila*, some studies provide evidence about ST, such as in *Drosophila* intestine [23]. The ST has also been extensively studied in *Drosophila melanogaster* with a modified P element, and it can cause damage in flies’ development [24]. Furthermore, the *white-peach* mutant is an excellent model for ST studies, where a *mariner* element has been inserted into the 5′-untranslated regulatory region of the *white* gene, resulting in a drastic reduction in the expression of this gene [25]. The *white* gene encodes a membrane protein responsible for transporting precursors of the drosopterin pigments that confer a reddish color to *Drosophila* eyes [26,27,28]. Excision of the *mariner* element of the *white* gene during development and prior to metamorphosis results in reversion of the *wpch* mutation in some of the cells that form the eye [29,30,31]. This is then reflected in the phenotype due to the appearance of mosaicism, where the eyes have some patches of the wild phenotype on the peach background (Figure 1).

Several studies have used the *wpch* mutant to characterize different aspects of *mariner* ST. Heat shock or even a rise in temperature is an activator of *mariner* ST [31,32,33,34]. However, other stressors, such as UV radiation, do not lead to a similar increase [34]. Somatic excision of *mariner* occurs during all developmental stages of the *Drosophila* lifecycle and is cumulative [35]. Using RT-qPCR (real-time quantitative PCR), it was estimated that in a strain carrying one autonomous copy of *mariner* and the non-autonomous *white-peach* copy, approximately 4% of the *mariner white-peach* copies are excised from the *white* locus in adult flies when the flies are maintained at 20 °C, and 7% when maintained at 28 °C [35]. This estimate suggests a high rate of somatic excision even in flies with low numbers of active elements.

However, it is important to emphasize that excision is only one step of TE transposition, and the final step of the transposition process is insertion. In this work, we developed a method to obtain insertion site sequences (ISS) and estimate the somatic insertion rate (SIR) of the *mariner* element in *Drosophila simulans*. We also analyzed long-read fragments to identify SM events. Our results suggest that even a few active copies of the *mariner* element can promote a high somatic insertion rate and the presence of SM events.

## 2. Materials and Methods

### 2.1. Drosophila Strain Used

The strain *Drosophila simulans white-peach* test (*wpcht*) was used. This isoline was produced [34] by crossing a *D. simulans white-peach* female (which had no active *mariner*-*Mos1*) [36] with wild-type *D. simulans* males collected in Brasília, Brazil (which had active *mariner*-*Mos1* elements). Furthermore, in the F2 generation, an isoline was established expressing the mosaic eye phenotype. This strain was estimated to have 1.84 (±0.83) *mariner* copies, the *mariner*-peach copy, which is not autonomous, and one autonomous copy, *mariner*-*Mos1* [34]. In our analyses, we did not differentiate autonomous or non-autonomous *mariner* copies and considered both of them.

### 2.2. Biological Treatments and DNA Extraction

We performed two different DNA extractions for data generation: one for short-read sequencing and the other for long-read sequencing (Figure 2). For molecular procedures for the short-read sequencing, flies were exposed to two different temperature conditions, at 20 and 28 °C, with three replicates at each temperature (Figure 2). Since 20 °C is the stocks temperature, we consider this condition as the control and 28 °C as the treatment. For the DNA extractions, 30 adult female *Drosophila* were used at 20 and 28 °C, from 1 to 4 days old. For extraction, the DNA purification kit (Ludwig Biotecnologia LTDA, Alvorada, Brazil) was used. Besides, we used the whole organism, without removing gonad/germline tissues. For the long-read sequencing, whole-genome DNA extraction of 20 flies of stocks at 20 °C was performed using the phenol-chloroform protocol [37] and using Oxford Nanopore Technology (GenOne, Rio de Janeiro, Brazil) (Figure 2).

### 2.3. Transposon Insertion Site Sequencing (TISseq)

We customized a method for sequencing the flanking regions of the *mariner* element and thus characterizing ISSs, called transposon insertion site sequencing (TISseq). The basic principle of this methodology is similar to the method described in [38]; however, it was developed and tested using a DNA element. For the preparation, whole genomic DNA was cleaved with two restriction enzymes, EcoRI and HindIII, which do not cut the *mariner* element and form cohesive ends. Adapters were annealed to DNA fragments, and PCR was used to amplify the upstream and downstream fragments of the *mariner* element, using primers bound to the subterminal regions of TE. Finally, the libraries were subjected to Miseq Illumina sequencing (Figure 1).

Briefly, for TISseq, genomic DNA was extracted from 30 flies (1 to 4 days old) for each replicate using a DNA purification kit (Ludwig Biotec, Alvorada, Brazil). DNA was cleaved with HindIII and EcoRI (Promega, Madison, WI, USA) according to the manufacturer’s instructions. Cleavage was performed at 37 °C for 4 h and enzymes were inactivated at 65 °C for 15 min. Fragments were ligated to the eco_cohesive adapter for EcoRI and hind_cohesive for HindIII using T4 DNA ligase (Invitrogen, Waltham, MA, USA). The adapters were phosphorylated with the enzyme T4 polynucleotide kinase (Promega, Madison, WI, USA) before ligation.

To isolate the flanking regions of *mariner* elements, cleaved genomic DNA ligated to adapters was amplified by PCR using primers annealing to the subterminal regions of the *mariner* element and primers corresponding to the adapters. The sequence of primers and adapters as well as a detailed molecular protocol can be found in Appendix A (TISseq molecular protocol).

The MiSeq Reagent Kit v2 (500 cycles) MS-102-2003 was used for sequencing, generating 10,000 reads for each sample (single-end). Six samples were sequenced, corresponding to three independent experiments. For each, two different treatments were performed, where the flies were kept at 20 or 28 °C. For each treatment, amplification was performed at both ends of the element, with two different PCRs being performed for each sample. Then, the two PCRs from the same sample in each of the replicates were merged and sequenced as the same sample.

### 2.4. Bioinformatics Analysis

Data analysis was performed using a pipeline written in the Bash and Python languages that was developed specifically for this study (Figure 3). First, sequence quality scores were generated using FastQC Read Quality Reports software (Galaxy version 0.72 + galaxy1). Then, a blind test was performed in which file names were scrambled to eliminate bias. Several external files were used during the pipeline, including files containing the sequences of the primer pairs used for the amplifications (complementary to the *mariner* element, both at the upstream and downstream ends), as well as the adapter sequence associated with the restriction enzyme site. These sequences were used because short sequencing can lead to artefacts and bias in analyses [39,40]. Here, we found some artefacts that were removed in the pipeline, such as false positives in the blast analysis. In this way, the pipeline was developed with trimming and other steps adapted to clean the known amplicon fragments—the adapters and the *mariner* region—and the generated artefacts. This increases the sensitivity and avoids false positives (Appendix A and Appendix A).

The pipeline is divided into four modules (Figure 3; see Appendix A for a detailed description of the pipeline). The first module is called “trimming” and consists of four steps: (1) only fragments with *mariner* sequences at one end are selected, (2) Fastq files are converted to fasta, (3) the adapter and adapter-chimaera sequences are trimmed—only fragments of interest, excluding adapters and primer sequences, are retained, and (4) sequences shorter than 25 nucleotides are removed. The second module (steps (5) and (6)) is the self-blast and output organization, in which sequences that have coverage and sequences that are observed only once are separated. Although some sequences lack coverage, they have the same potential as *mariner* insertion because they match the amplicon (adapter sequence plus interest sequence plus *mariner* and adapter sequence). The low or lack of coverage in these sequences is either due to the fact that we have a high SIR and the low sequenced reads are useful, or due to insertions that happened late during the development, and it was presented only in a few groups of cells, being minimally represented. Therefore, sequences with and without coverage were merged in the following steps. Steps 5 and 6 are performed with Blast tools (version 2.10.0+), using the same sample as query and subject. We used this alignment to identify sequences that were the same in one sample (repetitive sequences), i.e., the sequences of file “A” aligned with the same sequences of file “A”. This way, we can identify in the file “A” what are the sequences equal to or very similar inside of file “A”. In the third module, the redundant sequences are removed, a process divided into five steps: (7) Filter 1, which removes redundant in-coverage sequences. (8) Scov creation step to determine if the compared sequences are the same or not. Two parameters are defined: qCOV (query coverage) and sCOV (subject coverage). (9) The representative fragment between repeats was defined as sequences that had qCOV and/or sCOV values greater than or equal to 70. (10) The sequences with IDs defined as representative are retrieved, and (11) a final filter is performed to remove the repeated sequences that still remain in the samples. In the fourth module, the sequences without coverage are added to the representative sequences to be analyzed. First, these sequences are retrieved from the files of step (4) using the IDs generated in step (6) (step (12)). Then, these sequences are concatenated (step (13)) with the fasta sequences from step (11) to form a single file for each sample (step (13)), joining with and without coverage sequences. The fifth module concerns the comparison of insertions between treatments and consists of three steps. In the first step (14), the sequences of flies maintained at 20 °C are compared with those of flies grown at 28 °C, and vice versa. Then, filter 3 (step (15)) is performed to find any adapter variations, and finally, a reciprocal blast (step (16)) is performed to achieve greater specificity and sensitivity between alignments.

### 2.5. Somatic Insertion Rate Estimation (SIR)

A limitation of the sequencing strategy used (TISeq) is that fragments generated by ligation of primers located close to the *mariner* elements or short fragments lacking *mariner* are preferentially amplified by PCR and sequenced. For the Illumina kit used, the size of the sequenced fragments is about 300 bp. As a result, only a subset of insertions can be detected, namely those that have HindIII or EcoRI sites within approximately 300 bp of *mariner* elements. To estimate the maximum number of positions at which *mariner* can be inserted—that is, the scope of the method—we cleaved the genome of *D. simulans* (http://ftp.flybase.net/genomes/Drosophila_simulans/dsim_r2.02_FB2020_03/fasta/, accessed on 20 March 2021) in silico with both EcoRI and HindIII restriction enzymes. The fragments generated were tabulated and divided into fragments smaller and larger than 600 bp. The fragments smaller than 600 bp are those in which *mariner* insertions—at least in one end of *mariner*—can be completely detected by the method. For the larger fragments, only the insertions occurring within 300 bp of each end can be detected (Appendix A). The estimated genomic fraction in which the TISseq method can access the insertions is provided by Equation (1):(1)$$P=\frac{\;\left[\left(b\right)+\left(c\times 600\right)\right]\times 100}{\left(b\right)+\left(d\right)}$$
where *P* is the estimated genome portion which the TISseq methodology can access, *b* is the sum of bases of the fragments < 600 bp, *c* is the number of fragments > 600 bp, and *d* is the base sum of the fragments > 600 bp.

### 2.6. Whole-Genome Sequencing and Analyses

The sequence of the *mariner* element was recovered from NCBI and aligned to whole-genome sequencing using Blast (version 2.10.0+). There were no differences between autonomous and non-autonomous copies, with both being used in this analysis. To analyze the positions of the *mariner* element, its sequences approximately 2 kb upstream and downstream of the *mariner* element were recovered. The fragments were aligned in MEGAX using MUSCLE (Multiple Sequence Alignment) and the images and similarity table were created using CIAlign. Heatmap images were created using a custom pipeline in Google Collaboratory.

The different regions found upstream and downstream of *mariner* were recovered. The larger sequence of each region was used as a query, and these regions were searched in long-read sequencing. The recovered reads were analyzed in the same way as the *mariner*-containing fragments. This analysis can be used only if the studied region was unique in the genome. To verify this, the sequences used as queries were also aligned to the *Drosophila simulans* genome (http://ftp.flybase.net/genomes/Drosophila_simulans/dsim_r2.02_FB2020_03/fasta/, accessed on 20 March 2021), and if the region was unique, only one alignment was returned.

## 3. Results

### 3.1. ISS and SIR Estimation

We succeeded in obtaining the ISSs of the *mariner*, i.e., the flanking sequence of at least one side of the *mariner* position. However, more than half of the reads were discarded because of short lengths, sequencing chimeras, or amplified fragments that did not contain *mariner* (Appendix A). Figure 4 shows the number of ISSs from each experiment, upstream and downstream of the *mariner* insertions. We had little success attempting to pair upstream and downstream sequences at a single genomic position. This is because we only had short sequences, and fragments larger than 300 bp cannot be sequenced using this method (Appendix A). The comparison between the number of sequences with and without coverage showed that about 13% of the detected ISS were represented more than once, indicating low sequencing coverage. Nevertheless, some interesting results were obtained that deserve to be described.

A large number of ISSs were identified, ranging from 2494.6 to 1544.3 in the different treatments (Table 1). Most of the ISSs were represented as a unique read, and about 13% were represented with coverage, i.e., at least two reads cover the region. These results indicate that somatic insertions are a common phenomenon in this organism. For a preliminary estimate of the number of ISSs per fly, we first divided the number of ISSs detected by 30, which was the number of flies used to prepare each assay (sixth column in Table 1). This number ranged from 51.5 in the downstream region of flies maintained at 28 °C to 83.1 in the upstream regions of flies maintained at 20 °C. It is important to emphasize that we did not find a significant difference in ISS counts between treatments. When we compared the ratio between ISS and the number of trimmed sequences, the ratio was similar between treatments (Appendix A).

As discussed in the Materials and Methods Section, the TISseq method can only access TE insertions in which an EcoRI or HindIII site is located near (within approximately 300 bp) the *mariner* insertion (Appendix A). To estimate the portion of the genome accessed by TISseq, we performed an *in silico* cleavage of the *D. simulans* genome and obtained 78,455 fragments, of which 3048 were shorter than 50 bp and were excluded, 28,848 were between 50 and 600 bp, totaling 7,512,938 bp, and 49,605 were greater than 600 bp, totaling 29,763,000 bp. Thus, using the equation described in the Materials and Methods Section, we found that 29.8% of the genome was accessible by the TISseq method. The seventh column of Table 1 shows the ISS estimate, with the addition of the 70.2% that was presumed undetected. Thus, we estimate that the SIR has at least 173 to 279 somatic insertions of *mariner* for each adult fly shortly after emergence.

### 3.2. Analysis of Long-Read Fragments

The long-read fragments with low coverage were used to recover *mariner* elements and their insertions sites. A total of 20 *mariner*-containing fragments were found, and these elements were inserted at 4 different positions in the genome, underpinning the process of SM (Figure 5A). The peach copy was inserted into the *white* gene [25,29], promoting the *wpch* mutation. The location of *Mos1*, which was previously unknown, is probably in the histone cluster region (chromosome 2L) once 13 *mariner* elements were inserted and aligned with histone cluster genes. One *mariner* was inserted on chromosome 2R, three on chromosome X, and three on chromosome 3L. The results of the alignments can be found in Appendix A. The heatmap (Figure 5B) shows the sequence similarity outlining the four regions.

In order to show the SM events with long reads, besides *mariner*-containing fragments, the four regions upstream and downstream of *mariner* were found in the sequencing. The three regions were unique in the genome, and only histone regions (chromosome 2L) had multiple copies. There were fragments of the same region with and without *mariner* elements at chromosomes 2R and 3L, indicating the occurrence of SM of *mariner* in these regions. The event could be observed in upstream and downstream regions (Appendix A). The *white* gene could not show this result because reads were broken at the insertion region and the 2L chromosome region present in tandem and multiples copies, making the conclusion about excision or insertion impossible.

## 4. Discussion

Almost all studies on somatic transposition correlate this event with underlying diseases such as cancer [41] and Alzheimer’s disease [42], or ageing [43,44,45]. However, the expression [46] and excision [35] of TEs in normal somatic cells have already been described. In contrast, little is known about the somatic insertion of TEs, especially since this is one of the most important steps of the TS.

Treiber and Wadder [47] found no increase in somatic insertions with age in flies, and suggest that many of the somatic insertions are actually artifacts of sample preparation and sequencing. In our work, we have developed a sensitive methodology with specific trimming that considers the different possibilities of false positives. We have also confirmed SM events with long-read fragments showing the presence of *mariner* excision and insertion at different regions in the genome, even with low coverage.

Furthermore, Treiber and Waddell [47] suggest that somatic insertion is less common in *Drosophila* than previously thought. However, our data suggest otherwise. Based on the high SIR, approximately 13% of fragments had sequencing coverage. This value not only indicates that the sequencing plateau has not yet been reached, but also shows that SIR, although high, is still underestimated. In agreement with our results, the somatic excision rates already found for the same *D. simulans Wpcht* strain are high. About 4% of fly cells showed non-autonomous copy excision at 20 °C and about 7% of flies maintained at 28 °C [35]. Furthermore, somatic excision is cumulative throughout development and also occurs in adult individuals [35].

Considering the high somatic excision rate of *mariner* and the fact that it occurs cumulatively up to the adult stage [35], it is fitting that SIR was equally high. Although the excision rate was higher at 28 °C, with the large ISSs and sequencing coverage used, it was impossible to observe this difference in the insertion rate.

Siudeja et al. [23] present a measurement of somatic insertion of TEs into the genome of clonal cells in the intestine of tumor-bearing flies. The authors found a median of 15–23 somatic insertions per clonal genome, mainly from retrotransposons, with terminal inverted repeat (TIR) class elements being rare. Although this is one of the first works measuring somatic insertions, this study only looks at a specific tissue and detects insertions in clonal cell groups, which precludes the detection of unique somatic insertions. Furthermore, Siudeja et al. [23] point out that somatic insertions can be acquired during development as well as in adulthood. This event has already been proposed for *mariner* excision [34] and is now complemented by our data on insertion and long-read analyses.

Yu and collaborators [48] developed a method to measure the germinative insertion rate. Some work already indicates other methods for detecting insertions [23,38], improving these techniques [49], and detecting potential insertion sites [50]. However, these methods have not yet been developed and/or applied to the detection of whole organisms and DNA elements. Here, we illustrated the insertion rate of *mariner*, in a single generation of *D. simulans* considering the whole organism and only a few copies of the element in the genome.

## 5. Conclusions and Perspectives

Mobilization of TEs in somatic cells does not appear to be a rare event or caused only by stress. Our data showed that a low number of copies of the *mariner* transposon can promote a high rate of new somatic insertions throughout *Drosophila* development. We found that flies carrying two *mariner* copies (one autonomous and one non-autonomous) had on average 200 to 300 new somatic insertions at the moment the flies became adults. Although the SIR was high, it was an underestimate because the sequencing plateau was not reached in our study. Besides, TISseq is a promising method for measuring the ISS and SIR of different TEs in different organisms, with the only requirement being the sequence of the transposable element. In addition, it is believed that a high sequence coverage depth is required to measure ISS, as the SIR of *mariner*, for example, was much higher than previously thought. Finally, long-read sequencing allowed us to identify SM events even with low coverage, and we have been able to find *mariner* at different positions. Long-read sequencing can provide additional evidence of new insertion events as well as the flanking regions that may be under influence of the new inserted TE copy. Together, these approaches can help us to understand the SM process, especially the insertion process and its effects on genomes.

## Figures and Tables

**Figure 1 insects-13-00454-f001:**
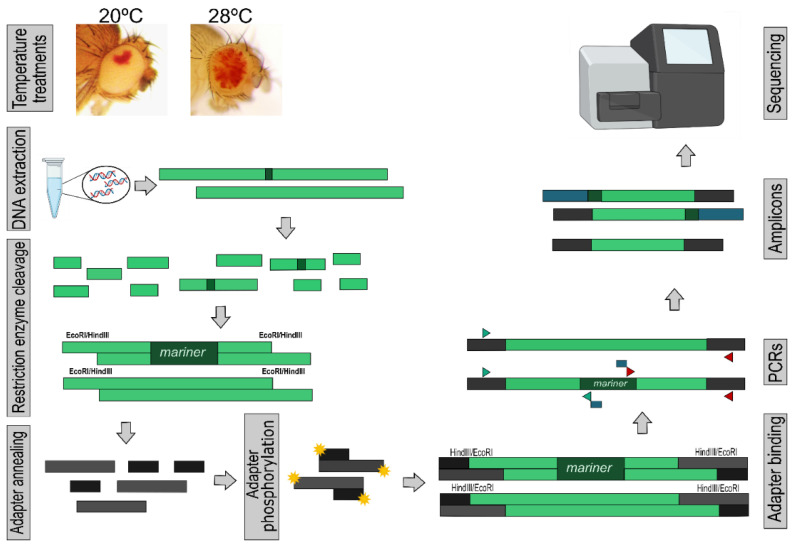
TISseq molecular protocol. This process used to produce the amplicons is described in detail and the steps are indicated by the light grey rectangles, namely: temperature treatments, DNA extraction, restriction enzyme cleavage, adapter annealing, adapter phosphorylation, adapter binding, PCRs, amplicons, and sequencing. The green rectangles represent DNA fragments, including the *mariner* element. The yellow star represents the phosphorylation process. There are two different adapters: darker grey rectangles represent one and the blue rectangles represent the other. Triangles represent the two primer pairs, as described in Appendix A. The blue rectangles represent binding at the mariner’s annealed primers and did not anneal at any sequence. They are included in the amplicons by the PCR process. There are three different amplicons being sequenced: two containing mariner (upstream and downstream) and one without mariner removed by bioinformatics analysis.

**Figure 2 insects-13-00454-f002:**
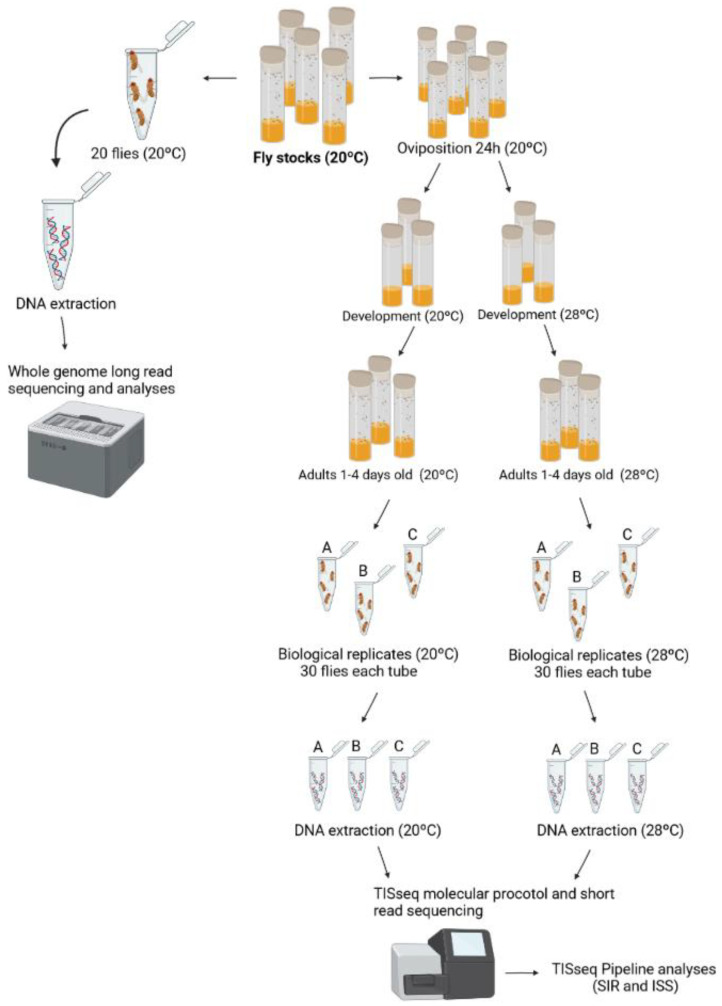
Biological treatments and experiment design. The fly stocks were maintained at 20 °C and the flies were used for DNA extraction and whole-genome sequencing, using long-read sequencing, and for treatment of temperature and short-read sequencing. First, 20 flies maintained at 20 °C were used for DNA extraction, and later, sequencing was performed. Second, mated flies (20 °C) were used for oviposition for 24 h at 20 °C. After that, half of the bottles were kept at 20 °C and the other half at 28 °C during the development. Flies that were born within 4 days (1–4 days) were collected and DNA was extracted, with three biological replicates at each temperature. The molecular protocol for the TISseq experiment was performed and short-read-sequenced. This figure was generated at BioRender.com (accessed on 20 March 2021).

**Figure 3 insects-13-00454-f003:**
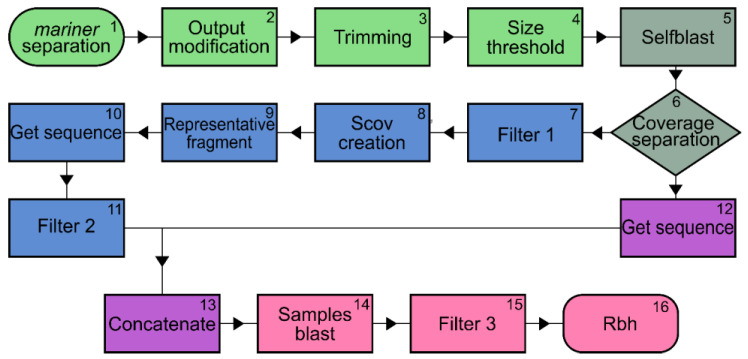
The flowchart demonstrates all steps of the data processing in steps 1 to 16. The colors indicate the five modules of the TISseq-Pipe pipeline, namely: Module 1: Trimming (green), Module 2: Self-blast and output organization (grey), Module 3: Removal of redundant sequences (blue), Module 4: Concatenated sequences (purple), and Module 5: Sample and reciprocal blast (pink).

**Figure 4 insects-13-00454-f004:**
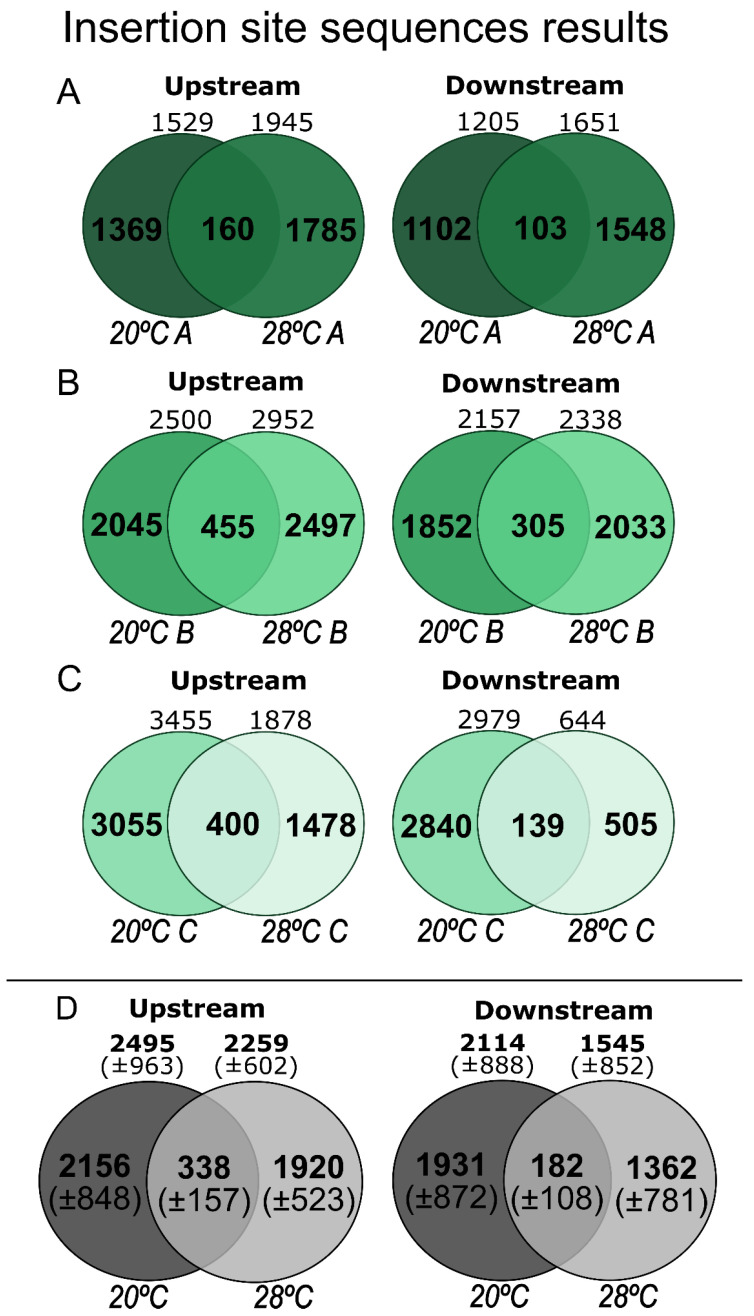
ISSs results. (**A**–**C**) Comparison between the 20 and 28 °C treatments in three replicates of the two ends of the element (upstream and downstream). Venn diagrams indicate the amount of sample fragments, what is unique to one or the other, and what is shared. (**D**) Means and standard deviations of the replicates.

**Figure 5 insects-13-00454-f005:**
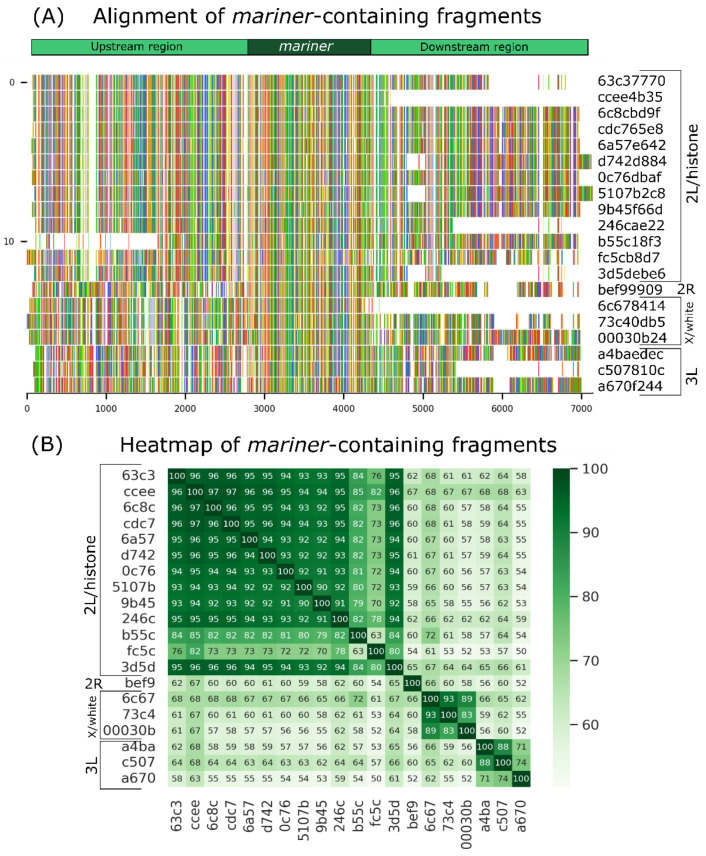
Alignment of *mariner*-containing long-read fragments and sequence similarity. (**A**) *mariner* is located in the central region of the figure, with 2 kb upstream and downstream of the element. The first 13 sequences are similar to each other and to chromosome 2L/*histone* genes: sequence 14 matches chromosome 2R, 15–17 match the X chromosome/*white* gene, and 18–20 match chromosome 3L. (**B**) Heatmap shows the similarity between sequences and delineates the most similar sequences in darker green. The 4 groups are highlighted, indicating the 4 different regions of the genome where the sequences aligned.

**Table 1 insects-13-00454-t001:** Somatic insertion values per genome and estimate for 100% of the genome. The means and standard deviations for each of the treatments and ends are represented according to the type of fragment and estimate.

	Fragments Trimmed (N)	Without Coverage	With Coverage	With + Without Coverage	Insertion Detected per Genome	Insertion Estimated per Genome
Upstream 20 °C	3533 ±1431.5	2164.6 ±780.7	330 ±183.9	2494.6 ±963.0	83.1 ±32.1	279.0 ±107.7
Upstream 28 °C	3155 ±790.5	1904.6 ±558.8	353.6 ±167.3	2258.3 ±601.6	75.3 ±20.0	252.6 ±67.3
Downstream 20 °C	2866.7 ±1186.6	1853.7 ±747.5	260 ±142.7	2113.6 ±887.8	70.4 ±29.6	236.4 ±99.3
Downstream 28 °C	1967.7 ±1043.9	1397 ±793.2	147.3 ±70.9	1544.3 ±852.0	51.5 ±28.4	172.7 ±95.3

## Data Availability

The data presented in this study are available in article or Appendix A.

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
