# Peer review of "Somatic Mobilization: High Somatic Insertion Rate of mariner Transposable Element in Drosophila simulans"

_insects, 2022, doi:10.3390/insects13050454_

Round 1
Reviewer 1 Report
The paper used an assay based on sequencing of the mariner flanking region and developed a pipeline to identify novel mariner insertions in Drosophila simulans at 20°C and 28°C. I think that this study is interesting on the somatic transposition and insertion of mariner transposon in Drosophila. There is, to date, no widely accepted methodological approach or strong guidelines for the analysis of somatic insertions in genomes. The present study developed pipeline and protocol, which consolidatesthe literature. However, this paper still needs some revision. Please refer the followings for detailed suggestions:
The title of the manuscript sounds more like conclusion. Suggest to revise and change the title if possible. Also, the study was conducted on a single species D. simulans, so should use species name instead of only the genus name Drosophila in title.
The authors have developed a pipeline for identification of mariner insertions which was tested in Drosophila simulans. Wouldn’t this MS belong to category of tools or software’s? Or was this pipeline designed for personal use only for the purpose of present study?
Introduction doesn’t provide enough information about somatic transposition specially in insects. I suggest to add a paragraph to introduce relative information about other insects or Drosophila species other than only D. simulans.
Lane 88: Provide information about collection of samples from where the white-peach strain was acquired. Also provide the overall %age or number of mariner elements of D. simulans.
Lane 90-91: not enough information. Suggest to add a table providing a detailed information about the flies exposed to the two environments i.e., number of flies, Stage of development e.g., Larvae, adult etc., and duration of exposures (in days).
Lane 91: Three replicates of each temperature was used. Did the author use a control group, too?
Lane 95-96: Provide al table containing information about the mariner TEs recovered from NCBI, like accession numbers, potentially active or not? Complete or truncated, length etc.
Lane 102: The mariner elements recovered was all the copies of the same element Mos 1? Please clarify this paragraph if the authors just targeted the autonomous Mos1 elements for analysis or all the mariner elements in the genome?
Lane 152-156: Provide the statistical result of the amplified fragments after sequencing in supplementary material 2 if available, i.e., Fragment length, the longest containing autonomous and non-autonomous elements, and the shortest fragments read.
Lane 164: “Only fragments with mariner sequences at one end are selected.” What about the fragment having mariner element in the middle and flanked by non TEs DNA at both end? Does the pipeline discard such sequences? If yes, I think that it will greatly impact the result and performance of the pipeline.
Lane 174: “Using the same sample as query and subject?” How did you do this? I didn’t get it, and would be hard for reader to understand. Briefly explain.
Table 1: Don’t need “Table with”, please revise the title. Column 1 header: What do you mean by ID? The column below representing temperature, upstream and downstream information. I suggest to rename it appropriately.
Lane 244-245: In silico analysis is not mentioned or explained in methodology. Or did I miss something? Please add in methods or delete here.
Lane 255: Were these all 20 elements, autonomous, non-autonomous, or both?
Lane 262 & fig.4b: “Heatmap shows the observe similarity between the aligned sequences.” Does it represent the similarity between mariner elements only? Or it shows similarity among the entire length of sequences including both up and downstream sequences? If that is the case, then how are most of the sequences more than 90% similar? Are these paralogous sequences or genes flanking the mariner elements? I am confused here. The sequences 63c37770, ccee4b35, 6c8cbd9f have 97% similarity, throughout the entire length? Or they are all flanked by histone sequences?
Conclusion is unnecessarily lengthy. Try to write a precise and to the point conclusion. Lane 332-334| is more like discussion, please revise.
Lane 335-337: I think you should delete the first line and rewrite the whole paragraph in a summarized sentence.
Description of the supplementary information is missing. Add a paragraph under heading, below the conclusion or anywhere else according to journal format.
Please check all the main text for the use of tense and punctuation mark, and revise. Also, for the references to be following the instruction of the journal.
Author Response
Dear reviewer,
We are very grateful for all reflections and suggestions on our work. The changes made in the text have been kept in red color, and the responses from rewire are listed below..
#Reviewer 1
#QUESTION1: The title of the manuscript sounds more like conclusion. Suggest to revise and change the title if possible. Also, the study was conducted on a single species D. simulans, so should use species name instead of only the genus name Drosophila in title.
#Answer: Altered in the text.
#QUESTION2: The authors have developed a pipeline for identification of mariner insertions which was tested in Drosophila simulans. Wouldn’t this MS belong to category of tools or software’s? Or was this pipeline designed for personal use only for the purpose of present study?
#Answer: The pipeline was designed thinking in all possible fault of this molecular experiment, using specific trimming to remove all adaptors and chimeras to evict interference in blast. Besides, the idea of pipeline is being tested and adapted to presented as a tool for Illumina short sequencing in other opportunity.
#QUESTION3: Introduction doesn’t provide enough information about somatic transposition specially in insects. I suggest to add a paragraph to introduce relative information about other insects or Drosophilaspecies other than only D. simulans.
#Answer: We have added some information about somatic mobilization of transposable elements in insects, but we have not found many papers measuring somatic mobilization, nor methods to do so..
#QUESTION4: Lane 88: Provide information about collection of samples from where the white-peach strain was acquired. Also provide the overall %age or number of mariner elements of D. simulans.
#Answer: Added in the text.
#QUESTION5: Lane 90-91: not enough information. Suggest to add a table providing a detailed information about the flies exposed to the two environments i.e., number of flies, Stage of development e.g., Larvae, adult etc., and duration of exposures (in days).
#Answer: More information was added in one new figure, explaining all designed methodology.
#QUESTION6: Lane 91: Three replicates of each temperature were used. Did the author use a control group, too?
#Answer: The treatment used consisted of only two different temperatures, and the flies are normally kept at 20ºC, so we consider the flies at 20ºC as the control group and those at 28ºC as the stress condition or treatment. Also, the same food and different characteristics were used for both groups. Eggs were laid at 20 ºC for 24 hours, after which half of the flasks were kept at 28 ºC and the other at 20 ºC. Flies were kept at this temperature throughout development until adult emergence and collected 1 to 4 days after emergence. One figure was created to better explain the treatment design.
#QUESTION7: Lane 95-96: Provide al table containing information about the mariner TEs recovered from NCBI, like accession numbers, potentially active or not? Complete or truncated, length etc.
#Answer: This information is presented in Supplementary material.
#QUESTION8: Lane 102: The mariner elements recovered was all the copies of the same element Mos 1? Please clarify this paragraph if the authors just targeted the autonomous Mos1 elements for analysis or all the mariner elements in the genome?
#Answer: The strain Drosophila simulans white-peach used in this work has around two copies of mariner, one is the autonomous copy mariner-mos1 and other the non-autonomous copy mariner-peach. For the analysis, both copies were recovered and analyzed without distinction. The difference between two copies is just 11 nucleotides (Jacobson and collaborators, 1986). This difference makes impossible to mariner-peach produced an active transposase, but it can use the mos1-transposase to move itself. In conclusion, the both elements can mobilize when mariner-Mos1 is presented and it is impossible differentiate the two copies in single fragments because of sequencing errors.
#QUESTION9: Lane 152-156: Provide the statistical result of the amplified fragments after sequencing in supplementary material 2 if available, i.e., Fragment length, the longest containing autonomous and non-autonomous elements, and the shortest fragments read.
#Answer: We provided information about the quantity of reads sequenced and possible to use in the Supplementary table 2. All the fragments of Illumina sequencing presented 305pb length. After trimming, they have different length. Because of this, we defined a threshold of 25pb and use fragments higher than 25pb in our analysis.
#QUESTION10: Lane 164: “Only fragments with mariner sequences at one end are selected.” What about the fragment having mariner element in the middle and flanked by non TEs DNA at both end? Does the pipeline discard such sequences? If yes, I think that it will greatly impact the result and performance of the pipeline.
#Answer: Fragments used in TISseq analysis are short fragments, about 300pb. Mariner elements are 1.3kb, so it's impossible to have mariner in the middle of the fragments. The restriction enzymes don't cleave marinerelement, only the surrounding regions. The mariner primers amplify the fragments between the end of the mariner element and the site of the restriction enzyme. This can be seen in Figure supplementary 2. We say that only fragments with mariner are selected because PCR can amplify fragments with adaptors at both ends (without mariner) and these fragments aren't mariner insertion regions.
#QUESTION11: Lane 174: “Using the same sample as query and subject?” How did you do this? I didn’t get it, and would be hard for reader to understand. Briefly explain.
#Answer: The blast was performed using the one sample aligned with itself. For example, we used the sequences of file “A” aligned with the same sequences of file “A”. This way, we can identify in the file “A” what are the sequences equals or very similar inside of file “A”. This method was used to identify in file “A” the repetitive sequences (same position in the genome that was represented more than one time - more than one sequence/read) and different sequences (position in the genome that has just one copy – only one sequence/read) and then, separate just one representant of each position.
#QUESTION12: Table 1: Don’t need “Table with”, please revise the title. Column 1 header: What do you mean by ID? The column below representing temperature, upstream and downstream information. I suggest to rename it appropriately.
#Answer: The header ID mean Identification of samples. It was replaced.
#QUESTION13: Lane 244-245: In silico analysis is not mentioned or explained in methodology. Or did I miss something? Please add in methods or delete here.
#Answer: In silico analysis was developed in methods part 2.5, to estimate somatic insertion rate (SIR) if we would can recover all the insertion with TISseq methodology. Follow: “To estimate the maximum number of positions at which mariner can be inserted - that is, the scope of the method - we cleaved the genome of D. simulans (http://ftp.flybase.net/genomes/Drosophila_simulans/dsim_r2.02_FB2020_03/fasta/) in silico with both EcoRI and HindIII restriction enzymes. The fragments generated were tabulated and divided into fragments smaller and larger than 600 bp.”
#QUESTION14: Lane 255: Were these all 20 elements, autonomous, non-autonomous, or both?
#Answer: We did not identify if these 20 mariner elements were autonomous or non-autonomous. The difference between the autonomous element (mariner-Mos1) and non-autonomous element (mariner-peach) is only 11 nucleotides, changing 4 amino acids. Because of low coverage and high error of long-read sequencing, we did not identify the elements. Both of them mobilize, if mariner-Mos1 is present.
#QUESTION15: Lane 262 & fig.4b: “Heatmap shows the observe similarity between the aligned sequences.” Does it represent the similarity between mariner elements only? Or it shows similarity among the entire length of sequences including both up and downstream sequences? If that is the case, then how are most of the sequences more than 90% similar? Are these paralogous sequences or genes flanking the mariner elements? I am confused here.
#Answer: It represents the similarity among mariner elements, upstream and downstream regions. We recovered all mariner elements presents in long-read sequencing (20 copies). Because of low coverage and no trimming has been performed, some positions have more than one representation in these 20 sequences. The mariner flanking by histone is overrepresented, probably because histone genes are presents in a lot of quantities in the genome. The other positions as 2R chromosome should be a new insertion in one group of cells of one fly. The white gene position, is a mariner-peach copy position. We did not know about Mos1 germline position. We believed that it is in histone/2L chromosome. The similarity is high because we imagine that positions were the same, of the same cell group, or a hotspot of mariner insertion, besides germ line positions.
#QUESTION16: The sequences 63c37770, ccee4b35, 6c8cbd9f have 97% similarity, throughout the entire length? Or they are all flanked by histone sequences?
#Answer: These three sequences have 97% throughout the entire length AND are flanked by histone sequences. The histone genes are overrepresented in this sequencing and highly conserved.
#QUESTION17: Conclusion is unnecessarily lengthy. Try to write a precise and to the point conclusion. Lane 332-334| is more like discussion, please revise.
#Answer: Altered in the text.
#QUESTION18: Lane 335-337: I think you should delete the first line and rewrite the whole paragraph in a summarized sentence.
#Answer: Altered in the text.
#QUESTION19: Description of the supplementary information is missing. Add a paragraph under heading, below the conclusion or anywhere else according to journal format.
#Answer: Added in the text.
#QUESTION20: Please check all the main text for the use of tense and punctuation mark, and revise. Also, for the references to be following the instruction of the journal.
#Answer: Added in the text.
Reviewer 2 Report
Cancian and collaborators developed a bioinformatic pipeline and performed experiments with a specific Drosophila strain that allow the quantification of a mariner transposable elements excision by a phenotypic assay. They reported interesting results about somatic excision and insertion (which was not characterized in this Drosophila lineage before) of the elements. Their results are timely and brings new light on TE somatic mobilization processes. However, I believe that more detailed description of the experimental conditions are needed to precisely compare the results from this methodology with phenotypic estimates of excision and evaluate the somatic insertion. Regarding this last point I think the most important information missing is if the authors dissected the flies and removed germline tissues or focused on tissues that are composed of only somatic cells. Follow below a number of suggestions.
Simple Summary: I suggest removing the word “Several”.
Page 1 line 14 - I suggest changing “D. simulans, and found “ to “D. simulans. We found “
Introduction:
Page 1 line 38 - In my opinion C. elegans is not the best example of the animal species with low TE content. I suggest that authors could quote other species with even lower TE content, for instance Anopheles darlingi.
Methods:
Page 3 line 96-109 - Why is this analysis section not into the bioinformatic pipeline description of page 4?
Page 3 line 102 - change “founded “ to found.
Do the authors remove gonad/germline tissues before DNA extraction?
How old were the flies at the extraction day. There is some information in page 4 line 124 (1 to 4 days old) but it is very short and vague. Why so many days between younger and old flies? The authors could standardize it sampling every individual that is born after 24h. Anyway, after 24 h is it enough to detect the omatid spots? Please clarify this section. In addition, I suggest the authors build a figure showing the experimental design (temperature, days until DNA extraction etc).
Did the authors estimate excision using the phenotypic excision assay (PMM)? I think it would be interesting to show some phenotypic estimate of excision using classical methodologies developed to whitepeach mutants.
Page 4 line 128 - “three independent experiments'' means three biological replicates for each temperature? The authors generated 10.000 ou 100.000 reads per sample? Here the experimental design figure I suggested above can help the reader understanding.
I could not find the supplementary material in the journal review page.
Figure 1 - Please describe in more detail triangles and rectangles of different colors meaning. Why are there some bluish rectangles in the final amplicon step?
Page 5 line 172 - The low coverage of many insertions may be a result of low representativeness of the new insertions (late insertion during the development of the fly which then only a few number of cell inherited new insertions) or low coverage sequencing. Both are probably contributing to singletons. The authors may add this additional explanation in the text.
Figure 2 - I suggest the authors use standard symbols for flowchart construction emphasizing the input and output of each step of the pipeline. https://en.wikipedia.org/wiki/Flowchart
Figure 3 - The overlapped region between the venn diagrams represents the number of Insertion site sequences that were exactly the same (100% identity) in the two experiments?
Results and Discussion
Page 6 line 228 - The authors stated “we first divided the number of ISSs detected by 30, which was the number of flies used to prepare each assay “ but in the material and methods section they described that the experiments were conducted with 20 flies “​​Whole-genome DNA extraction of 20 flies “. Please clarify.
Page 10 line 333 - Modify “high sequence coverage” to “ high sequence coverage depth”
Page 11 line 337 - I suggest rephrasing the sentence “Using long reads is another way to identify novel insertions” to “Long read sequencing can provide additional evidence of new insertion events as well as the flanking regions that may be under influence of the new inserted TE copy”
Author Response
Dear Reviewer,
We are very grateful for all reflections and suggestions on our work. The changes made in the text have been kept in red color, and the responses from rewire are listed below.
#Reviewer 2
#QUESTION1: Simple Summary: I suggest removing the word “Several”.
#Answer: Altered in the text.
#QUESTION2: Page 1 line 14 - I suggest changing “D. simulans, and found” to “D. simulans. We found”
#Answer: Altered in the text.
#QUESTION3: Introduction: Page 1 line 38 - In my opinion C. elegans is not the best example of the animal species with low TE content. I suggest that authors could quote other species with even lower TE content, for instance Anopheles darlingi.
#Answer: Altered in the text.
#QUESTION4: Methods: Page 3 line 96-109 - Why is this analysis section not into the bioinformatic pipeline description of page 4?
#Answer: We performed two different sequencing, the Illumina short read sequencing used in TISseq and the Pipeline to analyze; and the whole genome sequencing with long reads sequencing, to show, with another methodology, somatic insertions of mariner.
#QUESTION5: Page 3 line 102 - change “founded” to found.
#Answer: Altered in the text.
#QUESTION6: Do the authors remove gonad/germline tissues before DNA extraction?
#Answer: The gonad/germline were not removed. We understood and agreed that a little part of this insertion could be of germ insertion, and this point will be really considered in our future projects. However, we have a high proportion of somatic cells in relation of germ line cells. Besides, the previous work of Pereira and collaborators (2018) already suggest somatic excision of mariner element in almost all stages of development. So, if we have some insertions of germ line, we believed it should be little insertions in relation with somatic one.
#QUESTION7: How old were the flies at the extraction day. There is some information in page 4 line 124 (1 to 4 days old) but it is very short and vague. Why so many days between younger and old flies? The authors could standardize it sampling every individual that is born after 24h. Anyway, after 24 h is it enough to detect the omatid spots? Please clarify this section. In addition, I suggest the authors build a figure showing the experimental design (temperature, days until DNA extraction etc).
#Answer: To TISseq methodology we use low sample volumes in high concentrations. We started the cleavage with 1ug of DNA. Because of this, we beginning the experiment with 30 female flies each sample. We use flies from 1 to 4 days old to ensure that enough flies will be born during this period. The flies born with ommatid spots, although it is difficult to identified in the firsts hours after born because of flies are with lighter colors. Besides, we did not choose flies with spots to use in the experiments. All flies, until the necessary number, were used in the DNA extraction. One new figure was built with all designed used in the molecular experiments.
#QUESTION8: Did the authors estimate excision using the phenotypic excision assay (PMM)? I think it would be interesting to show some phenotypic estimate of excision using classical methodologies developed to whitepeach mutants.
#Answer: The excision assays were performed just in Pereira and collaborators (2018) to the same strain used in this work, here we just focus in the insertion analysis of mariner element.
#QUESTION9: Page 4 line 128 - “three independent experiments'' means three biological replicates for each temperature? The authors generated 10.000 ou 100.000 reads per sample? Here the experimental design figure I suggested above can help the reader understanding.
#Answer: We conduced three biological replicates for each temperature. The reads generated in each replicate and the reads used after trimming can be found in a detailed table in supplementary material.
#QUESTION10: I could not find the supplementary material in the journal review page.
#Answer: We will take care for this in the journal review page.
#QUESTION11: Figure 1 - Please describe in more detail triangles and rectangles of different colors meaning. Why are there some bluish rectangles in the final amplicon step?
#Answer: Additional information was detailed in the cation of figure. These bluish rectangles represent the other adapter sequence, and they did not anneal at any sequence. They are added in the amplicons by the PCR process.
#QUESTION12: There are two different types of adapters. The gray color adapter binding at the fragments by T4 DNA ligase and the blue color adapter that “bind” to fragment just after the amplification, because this adapter is connected in the primer. The blue color adapter can not anneal to the fragment, but the amplification introduced them in the fragment. More detailed will be add to the cation.
#Answer: Altered in the cation.
#QUESTION13: Page 5 line 172 - The low coverage of many insertions may be a result of low representativeness of the new insertions (late insertion during the development of the fly which then only a few number of cell inherited new insertions) or low coverage sequencing. Both are probably contributing to singletons. The authors may add this additional explanation in the text.
#Answer: Altered in the text.
#QUESTION14: Figure 2 - I suggest the authors use standard symbols for flowchart construction emphasizing the input and output of each step of the pipeline. https://en.wikipedia.org/wiki/Flowchart
#Answer: The figure was altered.
#QUESTION15: Figure 3 - The overlapped region between the venn diagrams represents the number of Insertion site sequences that were exactly the same (100% identity) in the two experiments?
#Answer: The overlapped region represents sequences with more than 90% of similarity, the major part with 100%. Besides, we just considered these sequences if they were presented in the reciprocal blast analysis, to confirm the presence of them in both analysis “file A versus file B” and “file B versus file A”.
#QUESTION16: Results and Discussion: Page 6 line 228 - The authors stated “we first divided the number of ISSs detected by 30, which was the number of flies used to prepare each assay “but in the material and methods section they described that the experiments were conducted with 20 flies “​​Whole-genome DNA extraction of 20 flies “. Please clarify.
#Answer: There are two different experiments and, so, two different extractions here. For the long-read sequencing (Oxford Nanopore sequencing) performance we used 20 flies for sample. For the short-read sequencing (Illumina sequencing), we used 30 flies for sample. This will be clarified in the new methodology figure.
#QUESTION17: Page 10 line 333 - Modify “high sequence coverage” to “ high sequence coverage depth”
#Answer: Altered in the text.
#QUESTION18: Page 11 line 337 - I suggest rephrasing the sentence “Using long reads is another way to identify novel insertions” to “Long read sequencing can provide additional evidence of new insertion events as well as the flanking regions that may be under influence of the new inserted TE copy”
#Answer: Altered in the text.
Reviewer 3 Report
The paper by Cancian et al. describes the activity of a small number of active mariner transposable elements in Drosophila simulans. The data presented show that even a small number of mariner elements can cause massive transposon insertions and excisions in somatic cells. The results are reminiscent of the data obtained by the Engels lab with P elements in Drosophila melanogaster.
Engels’ data showed that a single active, transposase-producing, P element could cause somatic lethality in the presence of 15 non-autonomous P elements.
The authors show using whole genome sequencing with Oxford Nanopore long-read sequencing that many novel 2kb flanking the novel insertions can used to accurately quantitate activity (Fig. 1), a method called TISeq. The authors also develop a bioinformatics pipeline to anayze the HTP genomic sequencing data. That should be useful for other groups.
The paper is appropriate for publication in Insects and should be of interest to anyone studying mobile genetic elements.
Author Response
Dear reviewer.
We are very grateful for all reflections and suggestions on our work. The changes made in the text have been kept in red color, and the responses from rewire are listed below..
#Rewier 3
Comments and Suggestions for Authors
The paper by Cancian et al. describes the activity of a small number of active mariner transposable elements in Drosophila simulans. The data presented show that even a small number of mariner elements can cause massive transposon insertions and excisions in somatic cells. The results are reminiscent of the data obtained by the Engels lab with P elements in Drosophila melanogaster. Engels’ data showed that a single active, transposase-producing, P element could cause somatic lethality in the presence of 15 non-autonomous P elements. The authors show using whole genome sequencing with Oxford Nanopore long-read sequencing that many novel 2kb flanking the novel insertions can used to accurately quantitate activity (Fig. 1), a method called TISeq. The authors also develop a bioinformatics pipeline to anayze the HTP genomic sequencing data. That should be useful for other groups. The paper is appropriate for publication in Insects and should be of interest to anyone studying mobile genetic elements.
#Answer: We are grateful for the understanding of the importance and consideration with our work and results.